# Multimodal LiDAR-Camera Novel View Synthesis with Unified Pose-free Neural Fields

**Weiyi Xue**[1,2], **Fan Lu**[1†], **Yunwei Zhu**[1], **Zehan Zheng**[1],
**Haiyun Wei**[1], **Sanqing Qu**[1], **Jiangtong Li**[1], **Ya Wu**[3], **Guang Chen**[1,2†]

{xwy, lufan, 2432040, zhengzehan, 2311399, sanqingqu, jiangtongli, guangchen}@tongji.edu.cn,
qpo144@163.com,
[1] Tongji University, [2] Shanghai Innovation Institute,
[3] CNNC Equipment Technology Research (Shanghai) Co., Ltd.
[†] Corresponding author

## Abstract

Pose-free Neural Radiance Field (NeRF) aims at novel view synthesis (NVS) without relying on accurate poses, exhibiting significant practical value. Image and LiDAR point cloud are two pivotal modalities in autonomous driving scenarios. While demonstrating impressive performance, single-modality pose-free NeRFs often suffer from local optima due to the limited geometric information provided by dense image textures or the sparse, textureless nature of point clouds. Although prior methods have explored the complementary strengths of both modalities, they have only leveraged inherently sparse point clouds for discrete, non-pixel-wise depth supervision, and are limited to NVS of images. As a result, a **M**ultimodal **U**nified **P**ose-free framework remains notably absent. In light of this, we propose **MUP**, a pose-free framework for LiDAR-Camera joint NVS in large-scale scenes. This unified framework enables continuous depth supervision for image reconstruction using LiDAR-Fields rather than discrete point clouds. By leveraging multimodal inputs, pose optimization receives gradients from the rendering loss of point cloud geometry and image texture, thereby alleviating the issue of local optima commonly encountered in single-modality pose-free tasks. Moreover, to further guide pose optimization of NeRF, we propose a multimodal geometric optimizer that leverages geometric relations from point clouds and photometric regularization from adjacent image frames. Besides, to alleviate the domain gap between modalities, we propose a multimodal-specific coarse-to-fine training approach for unified, compact reconstruction. Extensive experiments on KITTI-360 and NuScenes datasets demonstrate MUP's superiority in accomplishing geometry-aware, modality-consistent, and pose-free 3D reconstruction.

## 1 Introduction

Neural Radiance Fields (NeRFs) [20] have made substantial strides in novel view synthesis (NVS) for images and LiDAR point clouds [36, 53, 55, 12], with promising applications in autonomous driving scenarios [48, 39, 45, 46]. Recent developments have transitioned towards a pose-free paradigm [16, 10, 25, 3], facilitating reconstruction while accurately estimating sensor poses. This approach reduces dependence on time-consuming structure-from-motion algorithms like COLMAP [32] and on unreliable point cloud registration methods such as ICP [2, 29, 31, 28], both of which are susceptible to failure in wide-baseline scenarios.

However, existing pose-free NeRFs have largely concentrated on single modalities, particularly on images. Nevertheless, due to the lack of geometric consistency, relying solely on rich texture with-

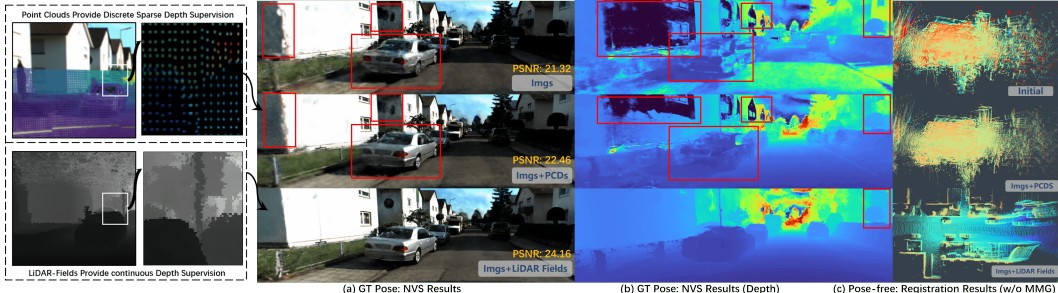

Figure 1: **NVS results w/ and w/o accurate poses.** Compared to continuous LiDAR-Camera Fields, projecting LiDAR point clouds onto images as discrete depth priors fails to provide continuous, pixel-wise supervision. Multimodal NeRFs (without MMG) leverage continuous LiDAR-Fields to constrain geometric consistency and optimize pose, aiding both reconstruction and pose optimization.

out geometry often leads to suboptimal results in large-scale scenes [19]. The assistance of Pseudo or real point clouds [3] can help address this issue. For instance, directly propagating depth loss to optimize geometric consistency [47] and leveraging depth for correspondence establishment and reprojection refinement [38, 3] can both contribute to pose optimization. Nonetheless, when performing pose-free reconstruction and projecting discrete point clouds onto images for depth supervision, only a sparse set of pixels contains depth information, underutilizing point cloud geometry. As shown in Fig. 1, LiDAR point clouds fail to provide pixel-wise supervision due to their inherent sparsity and discrete sampling. Comparing Fig. 1 reveals that supervision with discrete point clouds limits registration accuracy(c) and reconstruction quality(a), whereas continuous LiDAR-Field supervision excels in both. Moreover, Nope-NeRF [3] employs pixel-wise depth estimation to generate pseudo point clouds but suffers from scale ambiguity and limited accuracy. Consequently, even with point cloud [3, 35], image-based pose-free NeRFs [16, 19, 3] remain challenging to apply in large-scale scenes and are unable to perform point cloud NVS. Conversely, instead of utilizing point clouds as additional discrete supervision, recent advancements [36] have demonstrated that leveraging point clouds alone enables the reconstruction of continuous Neural LiDAR Fields. The LiDAR-based framework [55, 36, 12] facilitates highly accurate and continuous geometric reconstructions. Among these efforts, GeoNLF [47] extends LiDAR-Field to pose-free reconstruction task. Nonetheless, the inherent lack of texture information in point clouds, along with the sparse sampling and indistinct foreground-background boundaries in the range map, continues to constrain the performance.

Regarding the challenges encountered in the aforementioned single-modality approach, Continuous Neural LiDAR Fields can provide pixel-wise depth supervision for images and directly propagate gradients to pose estimation, providing continuous geometric constraints. In turn, images offer rich textures and clear boundaries, which enhance the registration accuracy of sparse point clouds. Consequently, reconstructing both point clouds and images as continuous neural fields allows them to effectively complement each other in pose-free scenarios. Nevertheless, prior research [37] has faced challenges due to the significant domain gap and uncoordinated convergence problems [27, 42, 34] between these modalities. Therefore, Alignmif [37] employs independent hash-grids for each modality. However, in pose-free, ill-conditioned optimization, jointly optimizing the two distinct hash grids and poses is infeasible and yields suboptimal performance compared to the single-modality model. The large discrepancy between two feature spaces leads to inconsistent gradients when propagated to poses, causing [37] to fail to converge.

Consequently, in pursuit of effectively integrating the two modalities for unified pose-free reconstruction, we introduce MUPa framework that facilitates the simultaneous reconstruction of both point clouds and images via a unified neural field. Specifically, to mitigate local minima issues in single-modality approaches, we propagate the gradient of multi-modal rendering loss to poses with varying emphasis at different optimization stages. Additionally, the MultiModal Geometric optimizer (MMG) guides pose optimization by leveraging geometric relations between multiview point clouds and incorporating point-to-image error as a regularization term. To alleviate modality conflicts [37] and address the uncoordinated convergence problem, we introduce a multimodal-specific coarse-to-fine training approach [16], facilitating the utilization of a singular hash grid for compact reconstruction. Moreover, to enhance color-depth consistency, we introduce a consistency constraint by projecting image pixels onto adjacent frames using depth derived from NeRF. Therefore, MUP

is capable of achieving geometry-aware, modality-consistent, and pose-free reconstruction in large-scale scenarios.

We evaluate our method across diverse scenarios using the KITTI-360 [15] and NuScenes [4] autonomous driving datasets. Comprehensive experiments demonstrate that MUP significantly outperforms prior state-of-the-art techniques and single-modality approaches by a large margin in both registration and NVS.

In summary, our primary contributions can be delineated as follows: (1) We propose MUP, a unified pose-free framework that combines the advantages of two modalities for pose estimation and multimodal NVS in large-scale scenes, efficiently leveraging a compact neural representation without the need for accurate poses. (2) We introduce a multimodal-specific training approach, integrated with the MMG module and consistency constraint, to facilitate modality-consistent, pose-free, and geometry-aware reconstruction. (3) We demonstrate the effectiveness of our method quantitatively and qualitatively through extensive experiments conducted on multiple datasets and scenes.

## 2 Related Work

**NeRF for Single-Modality NVS.** NeRF [20] and related works have achieved substantial progress in novel view synthesis. Diverse neural representations [21, 1, 5, 6, 11], techniques [22, 23, 41, 52], and generalization methods [7] for NeRF have been introduced to enhance its performance. Some methods incorporate depth priors [9, 30, 51] or point clouds as auxiliary data to ensure multi-view geometric consistency. However, relying solely on sparse depth supervision from point clouds fails to fully exploit their potential in expressing geometry. Consequently, researchers have extended the NeRFs to generate novel views from LiDAR [36, 12, 55, 53, 47], treating point clouds as range images. Nevertheless, sparse point clouds are notably deficient in dense texture information. Accordingly, we aim to leverage the complementary characteristics of both modalities, advancing a unified multimodal NeRF framework.

**Multimodal Joint Learning in NeRF.** NeRF framework facilitates the integration of a wide range of attributes into the volumetric rendering pipeline, including color [21], depth, intensity, and semantic labels [53]. Recent advancements [3, 9, 14, 30, 40] exploit point clouds to provide depth priors but fail to offer pixel-wise supervision. Neural sensor simulator Unisim [49] performs multimodal NVS via implicit fusion. However, all these methods rely on accurate poses. Very recently, Alignmif [37] has proposed using distinct hash grids for separate modality reconstruction followed by fusion. However, its intricate structure with two hash grids fails to converge in pose-free optimization and is highly computationally expensive. Our approach employs a more compact representation and introduces a novel strategy to achieve pose-free, multimodal reconstruction.

**NeRF with Pose Optimization.** Since iNeRF [50] and subsequent works [17, 8] demonstrated that NeRF can optimize the poses of new viewpoint images based on trained radiance fields, a series of approaches have aimed to reduce NeRF's reliance on highly accurate poses. NeRFmm [44] and SCNeRF [33] extend the method to intrinsic parameter estimation. BARF [16, 10] employs a coarse-to-fine reconstruction scheme that gradually learns positional encodings, demonstrating notable efficacy. Additionally, several studies have expanded BARF to tackle more challenging scenarios, such as sparse input [38], dynamic scenes [55, 18], and generalizable NeRF [7]. The coarse-to-fine training method has been particularly inspiring for our work. However, these pioneering efforts primarily target indoor or object-level scenes. Increasingly, pose-free methods have enhanced robustness by incorporating priors. In particular, [3, 38] use monocular depth or correspondence priors for constraints. Recently, [47] proposed a LiDAR-only pose-free framework. Nonetheless, the sparse nature of point clouds, coupled with the absence of texture information, ray-drop characteristics, and inherent noise, imposes limitations on registration accuracy. Moreover, all of the aforementioned methods are designed for a single modality. In the context of autonomous driving, they fail to fully exploit both the geometric information from point clouds and the texture information from images. As a result, a unified multimodal, pose-free framework remains absent.

## 3 Preliminaries

**Pose-free NeRF for Images and Point Clouds.** NeRF represents a 3D scene implicitly by encoding the density $\sigma$ along with additional data features such as color and intensity of the scene using an

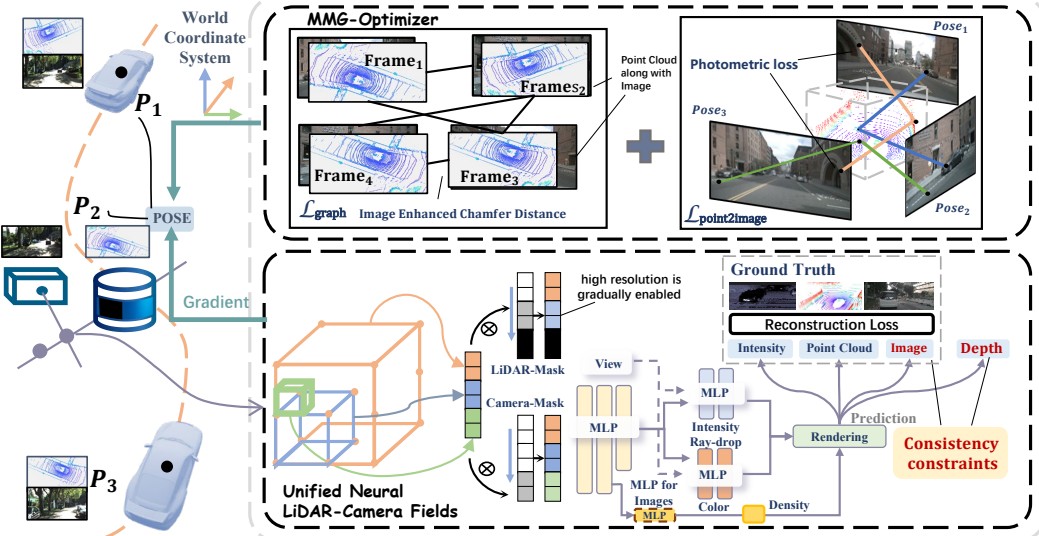

Figure 2: **Overview of our proposed MUP.** MUP derives pose gradients through both implicit global optimization from the Unified Neural LiDAR-Camera Fields and our explicit MMG-optimizer, both of which effectively leverage complementary multimodal information. Besides, we integrate Unified Neural LiDAR-Camera Fields with a multimodal-specific coarse-to-fine training strategy, along with consistency constraint to achieve geometry-aware, modality-consistent and pose-free reconstruction.

implicit neural function $F_\Theta(\boldsymbol{x}, \boldsymbol{d})$, where $\boldsymbol{x}$ is the 3D coordinates and $\boldsymbol{d}$ is the view direction. NeRF pipeline is compatible with both point clouds and images. For point clouds, it converts LiDAR point clouds into a range image, then casts a ray with a direction $\boldsymbol{d}$ determined by the azimuth angle $\theta$ and elevation angle $\phi$ under the polar coordinate system: $\mathbf{d} = (\cos\theta\cos\phi,\ \sin\theta\sin\phi,\ \cos\phi)^T$. When performing NVS, NeRF employs volume rendering techniques to accumulate densities and the pixel depth value $\hat{\mathcal{D}}$ along sampled rays. Using the same approach, NeRF predicts the color $\mathcal{C}$ for images or the intensity $\mathcal{S}$ and the ray-drop $\mathcal{R}$ for point clouds.

Traditional NeRF relies heavily on accurate sensor poses and reconstruction accuracy can be significantly compromised with imprecise poses. Pose-free NeRF is introduced to solve this issue by treating sensor poses $P = \{P_s | s = 0, 1...N - 1\}$ as optimizable parameters. Hence, the simultaneous update via gradient descent of $P$ and $\Theta$ can be achieved by minimizing $\mathcal{L} = \sum_{i=0}^{N} \|\hat{I}_i - I_i\|_2^2$ between the rendered and ground truth image or range image $\hat{I}, I$:

$$\Theta^*, \mathcal{P}^* = \arg\min_{\Theta, \mathcal{P}} \mathcal{L}(\hat{\mathcal{I}}, \hat{\mathcal{P}} \mid \mathcal{I}). \tag{1}$$

**Problem Formulation**. In large-scale autonomous driving scenarios, given time-synchronized sequences of cameras and LiDAR data, denoted as $\mathcal{I} = \{I_s | s = 0, 1, \ldots, N - 1\}$ and $\mathcal{Q} = \{Q_s | s = 0, 1, \ldots, N - 1\}$, the objective of MUP is to reconstruct the scene as a continuous implicit representation based on a unified neural field. MUP is capable of performing NVS for both modalities, while also simultaneously recovering the vehicle poses $P = \{P_s | s = 0, 1, \ldots, N - 1\}$, which enables the global alignment of both images and point clouds. The relative poses of all sensors with respect to the vehicle are assumed to be known.

**Pose Representation.** Following [3], pose is modeled as a rotation $\boldsymbol{R} \in SO(3)$ and a translation $\boldsymbol{t} \in \mathbb{R}^3$. This formulation allows for independent updates of the translation at the origin and the rotation around the origin. Rotation updates are computed in the Lie algebra of the special orthogonal group in three dimensions, $\boldsymbol{\phi} \in \mathfrak{so}(3)$, while translations are updated in $\mathbb{R}^3$. Specifically, the updates are expressed as $\boldsymbol{\phi}' = \boldsymbol{\phi} + \Delta\boldsymbol{\phi}$ and $\boldsymbol{t}' = \boldsymbol{t} + \Delta\boldsymbol{t}$. Here, $\boldsymbol{\phi}$ satisfies $\boldsymbol{R} = \sum_{n=0}^{\infty} \frac{1}{n!}(\boldsymbol{\phi}^\wedge)^n$, where $\boldsymbol{\phi}^\wedge$ denotes the skew-symmetric matrix representation of $\boldsymbol{\phi}$.

**Challenges in Multimodal NeRF.** Different modalities exhibit varying representations and converge at different rates [27, 42, 34]. Specifically, point clouds often converge faster than images due to their sparsity and direct geometric supervision. This disparity causes uncoordinated convergence, with the model shifting focus to images once the point cloud error is sufficiently small.

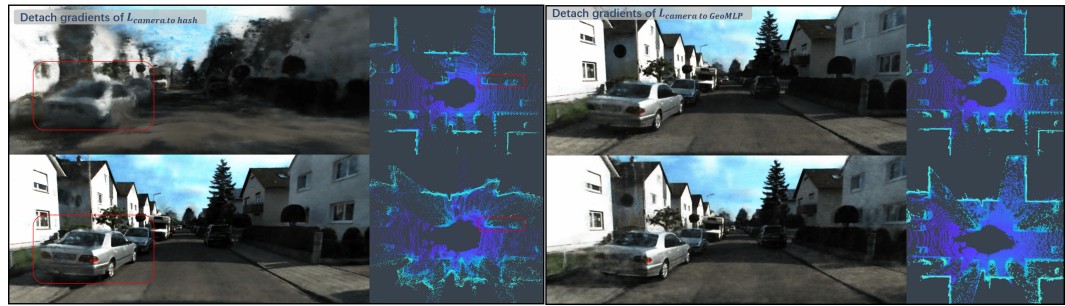

Figure 3: **Modality fusion in Hash-grids and geo-MLP.** We truncate the gradients of each modality separately in hash grids and geo-MLP. The results show that feature fusion across modalities primarily occurs in the hash grids rather than the geo-MLP.

## 4 Methodology

As shown in Fig. 2, our framework is divided into two main modules: a Unified neural fields that implicitly refines neural network and poses, and the Multimodal Geometric Optimizer (MMG) that explicitly optimizes poses. Both modules leverage geometric and texture information for registration and are executed alternately. In the following sections, we first present our LiDAR-Camera Fields in Section 4.1, integrating unified neural fields with a multimodal-specific coarse-to-fine training strategy for reconstruction and global implicit pose optimization. Then, we introduce our MMG module in Section 4.2, which provides explicit geometric guidance to avoid local optima. Finally, we present the proposed consistency constraint and the overall optimization pipeline in Section 4.3.

### 4.1 Unified Neural LiDAR-Camera Fields

To address the issue of unbalanced convergence speeds mentioned in Section 3, we design a Unified Neural LiDAR-Camera representation. Firstly, we introduce the Unified Neural Fields and identify that modality fusion occurs within the hash grids. Based on this observation, we propose a multimodal training method for optimizing the hash grid, which also stabilizes pose optimization and mitigates modality conflicts. Finally, a comprehensive analysis and discussion are provided.

**Unified Neural Fields.** Initially, we adopt i-NGP [21] as the base framework, leveraging multi-resolution hash grids to encapsulate the features, while a geometry-MLP (geo-MLP) is utilized to derive the density. In MUP, both the hash grids and the geo-MLP are shared across the modalities. For the image modality, *we use a lightweight MLP to refine the geo-MLP output, helping reduce modality conflicts.* To explore how modality features are fused, we independently truncate the gradients of reconstruction loss $L_{\text{Camera}}$ and $L_{\text{LiDAR}}$ to *hash grids* and *geo-MLP*. For hash grids, results indicate that truncating one modality prevents the Multimodal NeRF from learning the corresponding features. As shown in the upper images of Fig. 3, when the image gradient is truncated, novel views lose texture and resemble a colored point cloud projection, whereas truncating the point cloud gradient results in inaccurate geometry, resembling image-based pseudo point clouds. The same experiment on *geo-MLP* reveals slight performance degradation, suggesting feature fusion primarily occurs in the hash grids. *Thus, effectively controlling hash grid learning across modalities is crucial.*

**Multimodal-specific Coarse-to-fine Training.** To this end, we draw inspiration from the coarse-to-fine (C2F) strategy, which is widely used in pose-free NeRFs [47, 16, 10]. We extend this approach to a multimodal pose-free NeRF by adopting modality-specific C2F strategies, which helps to balance the influence of each modality on the hash grid. Specifically, we progressively activate shared hash grids from low to high resolution, employing distinct activation speeds and initiation points for each modality, as described in Eq. (2).

$$\gamma'^{\text{LiDAR}}_L = w_L(\alpha_{\text{LiDAR}})\gamma^{\text{LiDAR}}_L, \gamma'^{\text{Camera}}_L = w_L(\alpha_{\text{Camera}})\gamma^{\text{Camera}}_L, \tag{2}$$

where $\gamma'_L$ denotes the encoding of the $L$-th layer hash-grid, $w_L$ is the coarse-to-fine mask, and it can be any monotonic increasing function with a domain of $[L-1, L]$ and a range of $[0, 1]$, such as $w_L(\alpha) = \text{clip}(\alpha - L + 1, 0, 1)$ or $w_L(\alpha) = (1 - \cos(\text{clip}(\alpha - L + 1, 0, 1)\pi))/2$ in most pose-free methods [16, 47, 10]. $\alpha \in [0, L]$ is a controllable parameter proportional to the optimization progress. Notably, $\alpha$ varies across modalities. For images, all low-resolution hash grids are initially activated, with higher-resolution grids progressively activated. For the point cloud, a similar coarse-to-fine approach is used but with slower activation starting from low-resolution grids. In our experiment, the $\alpha_{\text{LiDAR}}$ is adjusted between 6-16, while the $\alpha_{\text{Camera}}$ is adjusted between 12-16.

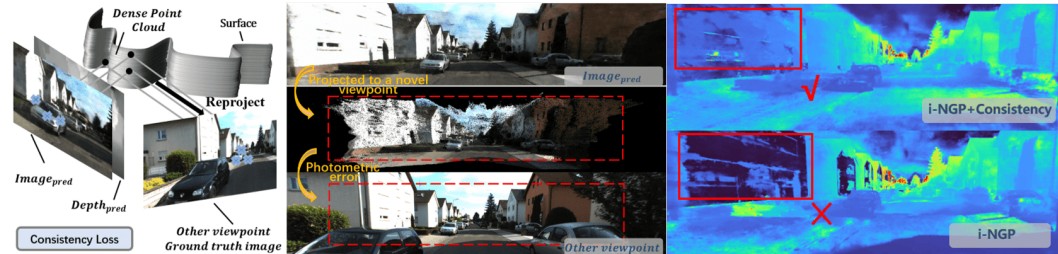

Figure 4: **Consistency constraint.** We project rendered images onto other frames by depth obtained from NeRF to compute the photometric error. It's particularly effective for textureless regions.

**Implicit Pose Optimization.** In the Unified NeRF training, gradients are also propagated to pose from reconstruction loss. However, in early optimization, geometric inaccuracies hinder texture-based pose refinement. In later stages, sparse point clouds limit registration accuracy, while images offer denser cues for alignment. Consequently, we adopt point cloud-based loss at the early stages and later employ image-based photometric loss to refine the poses. This is implemented by adjusting the learning rates of pose parameters across different modalities, as depicted in Eq. (3).

$$\mathbf{P}_{n+1} = \mathbf{P}_n - (1 - \mathtt{w})\mathtt{lr}G_{\text{LiDAR}} - \mathtt{w} \cdot \mathtt{lr}G_{\text{Camera}}, \tag{3}$$

where $G$ is the gradient of the corresponding modality, $\mathbf{P}_n$ denotes the pose at the $n$-th iteration, $\mathtt{w}$ is a control variable increasing progressively during the training process, $\mathtt{lr}$ denotes the learning rate.

**Discussion.** The C2F strategy is widely used in pose-free NeRFs [47, 16, 10]. During ill-conditioned optimization, minor perturbations in pose can lead to significant deviations in NeRF, potentially driving it towards a local minimum. The C2F strategy alleviates this issue by blocking partial gradient propagation, thereby mitigating the impact of such perturbations. The Jacobian of $\gamma'_L$ thus becomes:

$$\frac{\partial \gamma_L(\theta, \mathbf{x}; \alpha)}{\partial \theta} = w_L(\alpha)\frac{\partial \gamma_L(\theta, \mathbf{x})}{\partial \theta}, \; \frac{\partial \gamma_L(\theta, \mathbf{x}; \alpha)}{\partial \mathbf{x}} = w_L(\alpha)\frac{\partial \gamma_L(\theta, \mathbf{x})}{\partial \mathbf{x}} \tag{4}$$

where $\theta$ denotes the parameters of the hash grids, and the point $\mathbf{x}$ is associated with the pose. When $w_L(\alpha) = 0$, the contribution to the gradient from the $L$-th (and higher) resolution component is nullified. As shown in Eq. (4), the optimization of both hash-grid and pose follows a coarse-to-fine strategy. *In the early stages of optimization, only the gradients from the coarse resolution of the hash grids contribute to pose optimization, while the finer resolutions further refine the pose.*

Furthermore, our method utilizes a single shared hash grid and unified hash features across modalities. We adjusts the $\alpha$ for each modality, ensuring consistent convergence speeds and balanced loss across modalities. Moreover, the varying $\alpha$ values guide high-resolution hash grids to capture fine image textures, while the LiDAR field refines geometry and primarily supervises low-frequency geometric structures, mitigating cross-modal conflicts. In summary, our method ensures synchronous convergence and stable optimization, while also mitigating modality conflicts, by employing a distinct, compact, and efficient hash grid structure.

### 4.2 Multimodal Geometric Optimizer

Our MMG module leverages Image-enhanced Chamfer Distance combined with point-to-image regularization. Unlike implicit optimization through NeRF, MMG explicitly optimizes poses by leveraging both geometry and textures.

**Explicit Pose Optimization.** The closest-point correspondences between two partially overlapping point clouds establish the most direct geometric relationship, which can be leveraged effectively for registration like ICP [2]. Chamfer Distance (CD) is a well-established loss derived from point correspondences and can be computed as Eq. (5):

$$\mathbf{CD}_{(P,Q)} = \sum_{\mathbf{p}_i \in \mathbf{P}} w_i \min_{\mathbf{q}_i \in \mathbf{Q}} \|\mathbf{T}_\mathbf{P} p_i - \mathbf{T}_\mathbf{Q} q_i\|_2^2 + \sum_{\mathbf{q}_i \in \mathbf{Q}} w_i \min_{\mathbf{p}_i \in \mathbf{P}} \|\mathbf{T}_\mathbf{Q} q_i - \mathbf{T}_\mathbf{P} p_i\|_2^2, \tag{5}$$

where $q, p$ in point cloud $\mathbf{Q}, \mathbf{P}$ are homogeneous coordinates. $\mathbf{T}_P, \mathbf{T}_Q$ represent the transformation matrix to the world coordinate system. Additionally, we define $w_i$ to represent the weight of

Table 1: **Quantitative comparison of NVS in pose-free setting.** We conduct experiments under the pose-free setup. The estimated trajectory is aligned with the ground truth using Sim(3) for image-based methods. $\mathcal{PF}$: Pose-Free, $\mathcal{RR}$: Reconstruction after Registration, $\mathcal{I}$: Image-synthesizable, $\mathcal{PI}$: Image and Point cloud-synthesizable.

| Methods | Type | LiDAR Metrics | | | Image Metrics | | | Pose Metrics | | |
|---|---|---|---|---|---|---|---|---|---|---|
| | | CD↓ | F-score↑ | $\text{MAE}_I$↓ | PSNR↑ | SSIM↑ | LPIPS↓ | $\text{RPE}_t$ (cm)↓ | $\text{RPE}_r$ (deg) ↓ | ATE(m)↓ |
| Experiments on KITTI-360 [15] | | | | | | | | | | |
| Colored-ICP [24, 37] | $\mathcal{RR}./\mathcal{PI}$ | 0.492 | 0.787 | 0.149 | 20.92 | 0.698 | 0.459 | 25.383 | 0.899 | 1.624 |
| Nope-NeRF [3, 56] | $\mathcal{PF}./\mathcal{I}$ | - | - | - | 19.82 | 0.337 | 0.592 | 83.223 | 14.412 | 0.653 |
| BA-Alignmif [37, 10] | $\mathcal{PF}./\mathcal{PI}$ | 0.641 | 0.722 | 0.116 | 19.12 | 0.641 | 0.439 | 36.179 | 0.498 | 0.410 |
| MUP(Ours) | $\mathcal{PF}./\mathcal{PI}$ | 0.079 | 0.942 | 0.096 | 23.46 | 0.759 | 0.287 | 1.471 | 0.025 | 0.187 |
| Experiments on NuScenes [4] | | | | | | | | | | |
| Colored-ICP [24, 37] | $\mathcal{RR}./\mathcal{PI}$ | 0.930 | 0.599 | 0.047 | 19.21 | 0.438 | 0.644 | 14.380 | 0.599 | 1.170 |
| Nope-NeRF [3, 56] | $\mathcal{PF}./\mathcal{I}$ | - | - | - | 18.01 | 0.341 | 0.670 | 129.899 | 12.399 | 0.718 |
| BA-Alignmif [37, 10] | $\mathcal{PF}./\mathcal{PI}$ | 1.695 | 0.603 | 0.044 | 18.73 | 0.621 | 0.619 | 182.391 | 0.377 | 4.266 |
| MUP(Ours) | $\mathcal{PF}./\mathcal{PI}$ | 0.810 | 0.656 | 0.042 | 20.83 | 0.699 | 0.585 | 4.058 | 0.101 | 0.176 |

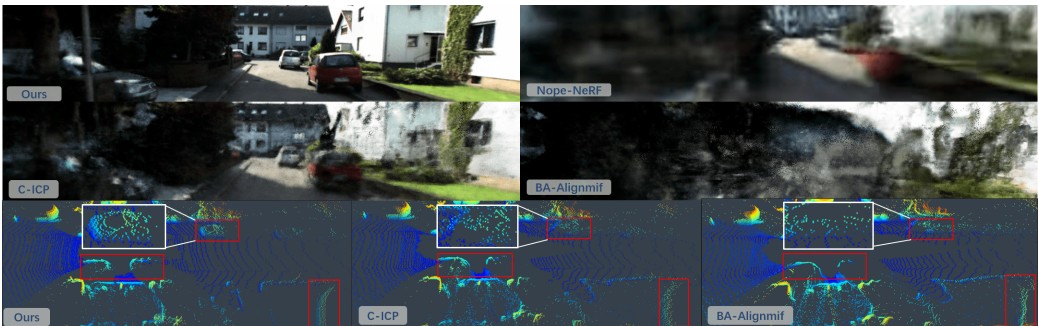

Figure 5: **Qualitative comparison of NVS.** We compared MUP with pose-free and registration-first methods. Nope-NeRF and Colored-ICP-assisted fail due to the large-scale scene. BA-Alignmif struggles to converge. All baselines fail entirely on certain sequences.

each correspondence. Our MMG module directly computes the inter-frame CD and propagates the gradient to the poses, guiding the optimization of the poses during the ill-conditioned optimization. However, CD overlooks the partial overlap of point clouds, merely minimizing CD does not necessarily improve pose accuracy. Leveraging the multimodal input, we exploit images to alleviate the impact of non-overlapping regions. Specifically, we project point clouds onto time-synchronized images to derive color information. During the calculation of CD, we re-weight the relevant correspondences by incorporating photometric information.

$$w_i = \begin{cases} 1 & \|C\langle\mathcal{F}(p_i)\rangle - C\langle\mathcal{F}(q_i)\rangle\|_1 \le m \\ 0 & \|C\langle\mathcal{F}(p_i)\rangle - C\langle\mathcal{F}(q_i)\rangle\|_1 > m, \end{cases} \tag{6}$$

where the $C\langle\cdot\rangle$ denotes the color obtained by sampling from the image. $\mathcal{F}(x)$ denotes the projection function that maps a 3D point $x$ onto a 2D image. Additionally, following [47], we further construct a multi-frame graph $(\mathcal{S}, \mathcal{E})$, where vertex $\mathcal{S}$ represents a frame of point cloud along with the time-synchronized image, each edge $\mathcal{E}$ represents the CD. Finally, the Image-enhanced CD is computed as $\mathcal{L}_{\text{ICD}} = \sum_{(i,j)\in\mathcal{E}} \mathbf{CD}_{(i,j)}$.

**Point-to-Image Regularization.** As shown in Fig. 2 (top right), we project point clouds onto images from adjacent frames to establish pixel correspondences. Based on these correspondences, we introduce a point-to-image error using photometric loss, which serves as a regularization term.

$$\mathcal{L}_{\text{Point2Image}} = \sum_{ij}\|C_i\langle\mathcal{F}_i(P_i)\rangle - C_j\langle\mathcal{F}_j(T_jT_i^{-1}P_i)\rangle\|_2^2, \tag{7}$$

where $P$ denotes the point cloud. Finally, the overall optimization loss for MMG is defined as:

$$\mathcal{L}_{\text{MMG}} = \mathcal{L}_{\text{IRCD}} + \mathcal{L}_{\text{Point2Image}}. \tag{8}$$

### 4.3 Overall Optimization Pipeline

**Consistency Constraint.** Due to the differing Fields of View (FoV) between LiDAR and cameras, plain LiDAR-Fields provide incomplete and limited depth supervision. Therefore, we further introduce a geometric consistency constraint, which leverages reprojection error to constrain regions outside the LiDAR's FoV. Specifically, we extract dense point clouds from rendered depth maps $z$ and project them onto adjacent frame images to compute photometric errors. It is effective for large textureless regions, enforcing geometric-color consistency, and is calculated as Eq. (9):

$$\mathcal{L}_{\text{cons}} = \sum_{(i,j)} \|C_i \langle p \rangle) - C_j \langle \mathcal{F}_j(T_j T_i^{-1} \mathcal{F}^{-1}(z,p)) \rangle\|_2^2, \tag{9}$$

where $\mathcal{F}^{-1}(z,p)$ denotes the back-projection, mapping a pixel $p$ to a 3D point using depth $z$.

**Optimization.** To optimize MUP, the total reconstruction loss is formulated as a weighted sum of intensity loss $\mathcal{L}_S$, ray-drop loss $\mathcal{L}_R$, LiDAR range image loss $\mathcal{L}_D$, photometric loss $\mathcal{L}_{rgb}$ from the image and consistency loss $\mathcal{L}_{\text{cons}}$:

$$\mathcal{L}_{\text{D}}(\mathbf{r}) = \sum_{\mathbf{r} \in R} \|\hat{D}(\mathbf{r}) - D(\mathbf{r})\|_1 , \mathcal{L}_{\text{rgb}}(\mathbf{r}) = \sum_{\mathbf{r} \in R} \|\hat{I}(\mathbf{r}) - I(\mathbf{r})\|_2^2 \tag{10}$$

$$\mathcal{L}_{\text{S}}(\mathbf{r}) = \sum_{\mathbf{r} \in R} \|\hat{S}(\mathbf{r}) - S(\mathbf{r})\|_2^2 , \mathcal{L}_{\text{R}}(\mathbf{r}) = \sum_{\mathbf{r} \in R} \|\hat{R}(\mathbf{r}) - R(\mathbf{r})\|_2^2 \tag{11}$$

$$\mathcal{L} = \lambda_\alpha \mathcal{L}_{\text{D}} + \lambda_\beta \mathcal{L}_{\text{rgb}} + \lambda_\gamma \mathcal{L}_{\text{S}} + \lambda_\eta \mathcal{L}_{\text{R}} + \lambda_{\text{r}} \mathcal{L}_{\text{cons}} \tag{12}$$

## 5 Experiment

### 5.1 Experimental Setup

**Datasets and Experimental Settings.** We conducted experiments on two public autonomous driving datasets: NuScenes [4] and KITTI-360 [15] dataset, each with five representative time-synchronized LiDAR point cloud and image sequences. For the NuScenes dataset, it includes six cameras and a LiDAR sensor, with keyframes that are typically used, which are time-synchronized based on timestamps. Following [47] we selected 33 consecutive frames from keyframes as a single scene. KITTI-360 has three cameras and a LiDAR, where each frame's point cloud and image are time-aligned. Following [36, 55, 37], we use the standard KITTI-360 dataset, all images and point clouds are time-synchronized. For both datasets, only the front-facing single camera was utilized. Following [47, 16], we perturbed poses of car with additive noise corresponding to a standard deviation of $20 \deg$ in rotation and $3m$ in translation. The relative poses of all sensors with respect to the vehicle are assumed to be provided.

**Metrics.** We evaluate our method for pose estimation and NVS. For pose estimation, we follow [3], employing standard odometry metrics: Absolute Trajectory Error (ATE) and Relative Pose Error (RPE), with rotational (RPE$_r$) and translational (RPE$_t$) components. Following [36, 55] for point cloud NVS, we adopt CD to assess 3D geometric errors and the F-score with a 5 cm threshold. We also compute mean absolute error (MAE) for intensity in projected range images. Besides, we follow the approach in [3, 37], employing PSNR, LPIPS [54], and SSIM [43] for image NVS.

**Implementation Details.** All experiments were conducted on a single NVIDIA GeForce RTX 3090 GPU. 768 points were uniformly sampled along each ray for two modalities. MUP optimization was implemented in PyTorch [26] using the Adam optimizer [13]. The learning rates were set as follows: $1 \times 10^{-2}$, decaying to $1 \times 10^{-4}$ for NeRF; $1 \times 10^{-3}$, decaying to $1 \times 10^{-5}$ for translation; and $5 \times 10^{-3}$, decaying to $5 \times 10^{-5}$ for rotation. The weighting coefficients for each loss term are defined as: $\lambda_\alpha = \lambda_\beta = 1000, \quad \lambda_\gamma = 10, \quad \lambda_\eta = 2.5, \quad \lambda_r = 150$. Besides, all sequences are trained for 60K iterations in the pose-free setting and 30K iterations when ground truth poses are available. Additionally, after every $m_1$ epoch of Unified NeRF training, we proceed with $m_2$ epochs of pure geometric optimization, where the ratio $m_2/m_1$ decreases from 10 to 1. Both $\alpha$ values are tuned so that the coarse-to-fine strategy is applied during the training progress between 0 and 0.3. In the training process, w increases from 0 to 1.

| Methods(Pose - free) | LiDAR | | Image | Pose Metrics | | |
|---|---|---|---|---|---|---|
| | CD↓ | F-score↑ | PSNR↑ | RPE$_t$ ↓ | RPE$_r$ ↓ | ATE↓ |
| w/o MMG | 0.592 | 0.731 | 19.27 | 26.201 | 0.433 | 0.805 |
| w/o $\mathcal{P}2\mathcal{IR}$ | 0.083 | 0.936 | 23.35 | 1.533 | 0.041 | 0.205 |
| w/o Image | 0.089 | 0.937 | - | 1.668 | 0.062 | 0.224 |
| w/o MSC2F | 0.113 | 0.932 | 22.06 | 1.542 | 0.058 | 0.256 |
| MUP(Ours) | 0.079 | 0.942 | 23.46 | 1.471 | 0.025 | 0.187 |

| Methods(GT - pose) | LiDAR Metrics | | | Image Metrics | | |
|---|---|---|---|---|---|---|
| | CD↓ | F-score↑ | MAE$_I$↓ | PSNR↑ | SSIM↑ | LPIPS↓ |
| w/o Cons | 0.092 | 0.931 | 0.096 | 23.66 | 0.793 | 0.227 |
| w/o MSC2F | 0.113 | 0.923 | 0.102 | 23.24 | 0.798 | 0.230 |
| MUP(Ours) | 0.080 | 0.945 | 0.089 | 24.29 | 0.812 | 0.211 |

Table 2: **Ablation studies on the MMG module and image modality under the pose-free setting(top).** MMG module plays a pivotal role in pose optimization. **Ablations of MSC2F and consistency constraint under GT-pose setting(bottom).** $\mathcal{P}2\mathcal{IR}$: Point2Image Regularization. MSC2F: Training Strategy.

| Methods | LiDAR Metrics | | | Image Metrics | | |
|---|---|---|---|---|---|---|
| | CD ↓ | F-score ↑ | MAE$_I$ ↓ | PSNR ↑ | SSIM ↑ | LPIPS ↓ |
| Experiments on KITTI - 360 [15], ī-NGP: i-NGP w/ point cloud. | | | | | | |
| i-NGP [21] | - | - | - | 23.12 | 0.791 | 0.223 |
| L-NeRF [36] | 0.083 | 0.942 | 0.097 | - | - | - |
| ī-NGP [21] | - | - | - | 23.23 | 0.794 | 0.220 |
| MUP(Ours) | 0.080 | 0.945 | 0.089 | 24.29 | 0.812 | 0.211 |
| Experiments on NuScenes [4]   , ī-NGP: i-NGP w/ point cloud. | | | | | | |
| i-NGP [21] | - | - | - | 20.78 | 0.667 | 0.530 |
| L-NeRF [36] | 0.815 | 0.673 | 0.041 | - | - | - |
| ī-NGP [21] | - | - | - | 20.92 | 0.682 | 0.564 |
| MUP(Ours) | 0.798 | 0.678 | 0.038 | 21.53 | 0.704 | 0.545 |

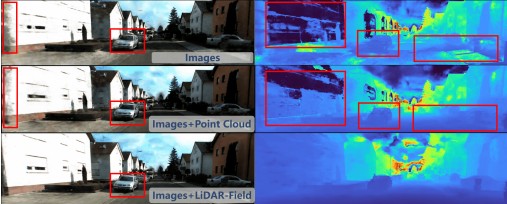

Table 3: **Quantitative comparison on NVS with GT-poses.** We conducted experiments under GT-poses to demonstrate the effectiveness of our method in modal fusion.

Figure 6: **Ablation in pose-free setting.** The first row illustrates registration results w/ and w/o the MMG, while the second row compares depth maps w/o and w/o the image modality.

Figure 7: **Qualitative NVS results with GT-poses.** MUP outperforms single-modal methods i-NGP w/ and w/o point clouds and LiDAR-NeRF. Our method achieves significantly better depth estimation and NVS quality.

## 5.2 Comparison and Ablation in Pose-free Setting

**Baselines.** In pose-free settings, all baselines utilize multimodal inputs. Methods are divided into two distinct categories: one where registration is performed prior to reconstruction, and another where pose-free reconstruction simultaneously estimates the poses. Following prior pose-free NVS studies [16, 19, 47, 3, 10, 44], reconstruction is typically performed on short sequences without real-time constraints. Consequently, incremental SLAM systems are not directly comparable, and directly frame-to-frame registration is adopted instead [47].

For the first category, the registration method Colored-ICP [24] integrates both point clouds and images. We utilize it for registration and subsequently apply Alignmif [37] for multimodal reconstruction. Notably, For the second category, Nope-NeRF takes images input and leverages DPT [56] to construct pseudo point clouds. Moreover, we reformulate the Alignmif method in a pose-free framework, allowing gradient propagation to the poses for optimization. For all pose-free methods, we adopt the strategy from [44, 47] to obtain the poses of test views for rendering.

**Comparison in NVS and Pose Estimation.** The quantitative and qualitative results are presented in Table 1 and Fig. 5. Our method outperforms all approaches in both modalities. Previous pose-free methods Nope-NeRF [3] primarily targeted small-scale scenarios, and only can perform images NVS. Consequently, when applied to autonomous driving environments, Nope-NeRF underperforms due to the lack of scale and the imprecision in depth estimation. Alignmif [37] cannot be effectively used in ill-conditioned optimization. Its complex structure with multiple independent hash grids prevents efficient registration and reconstruction. As for the registration-first approach, Colored-ICP [24] exhibits limited accuracy in large-scale outdoor scenes. Our method achieves the highest pose estimation accuracy.

**Ablation Study in pose-free setting.** All ablation studies are conducted on KITTI-360 [15]. We firstly exclude the images and perform reconstruction using only the LiDAR point clouds. Table 2 underscores the critical role of multimodal fusion in enhancing accuracy. Besides, as shown in Fig. 6, the incorporation of images mitigates the limited LiDAR field of view, thereby enabling the

acquisition of depth maps (or point clouds) with a broader FoV. It also introduces complementary information beyond LiDARs perspective, enhancing registration accuracy. We also conduct ablation studies on MMG module. The results indicate that relying solely on NeRF's implicit pose optimization fails to achieve accurate pose estimates and leads to convergence at local optima. Besides, we conduct ablation studies on image enhancement in MMG and the modality-specific C2F training strategy (MSC2F). All modules demonstrate effectiveness in pose-free experiments.

### 5.3 Ablation Study with Ground Truth Poses

In the pose-free setting, the precision of the estimated poses and the efficacy of our MMG module are pivotal to performance. Therefore, we exclude the MMG and conduct ablation experiments with GT pose to further demonstrate the advantages of our Unified NeRF with the multimodal-specific coarse-to-fine training strategy (MSC2F) and the consistency constraint in multimodal fusion. Additionally, to further demonstrate the effectiveness of our multimodal approach, We conduct comparative experiments with the single-modality LiDAR-NeRF [36] and i-NGP [21], where i-NGP is tested both with and without utilizing discrete LiDAR point clouds as depth supervision.

**Ablation of MSC2F and Consistency Loss.** Table 2 presents the ablation results on MSC2F and consistency constraint under the GT-Pose setting to verify the effectiveness of our method. By using a reprojection operation to link geometry and color, our method effectively ensures geometry-color coherence, resulting in improved reconstruction quality in both the image and point cloud NVS.

**Comparision with Single-modality Methods.** The quantitative and qualitative results are presented in Table 3 and Fig. 7. Our MSC2F fusion approach, along with the color-depth consistency constraint, effectively integrates features from both modalities. Thus, compared to single-modality methods and i-NGP [21] that with and without point clouds for depth supervision, we achieve high-quality NVS and the best results across both modalities.

## 6 Limitation

MUP demonstrates strong performance in pose-free multimodal NVS and pose estimation under challenging large-scale scenes. However, it is primarily designed for sensor data within a sequence and relies on temporal correlations between frames. Additionally, it is not designed to handle dynamic scenes, which is a non-negligible limitation in autonomous driving scenarios.

## 7 Conclusion

We revisit the limitations of single-modality pose-free methods in large-scale scenes. Subsequently, we propose a novel framework for Multimodal Unified Pose-free LiDAR-Camera NVS. Benefiting from the unified neural representation with MSC2F training strategy, the color-depth consistency constraint, the MMG module, and most importantly, the integration of different modalities and our pose optimization approach, we achieve geometry-aware, modality-consistent, pose-free reconstruction.

## 8 Acknowledgement

This work was supported by the National Key Research and Development Program of China (No. 2024YFE0211000), in part by the National Natural Science Foundation of China (No. 62372329, 62402341), in part by the Shanghai Scientific Innovation Foundation (No. 23DZ1203400), in part by the China Postdoctoral Science Foundation (No. BX20250383, GZB20250385, 2025M771530, 2025M771539, GZC20241225, 2025M771513), in part by Tongji-Qomolo Autonomous Driving Commercial Vehicle Joint Lab Project, and in part by Xiaomi Young Talents Program.

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
