# OpenReview forum: "Multimodal LiDAR-Camera Novel View Synthesis with Unified  Pose-free  Neural  Fields"
_NeurIPS.cc/2025/Conference — NeurIPS 2025 poster_

### Official Review · Reviewer_RGqZ · 2025-06-11

**Clarity:** 2
**Significance:** 2
**Originality:** 3
**Rating:** 4
**Confidence:** 3

**Summary:**

This paper proposes a novel unified filed for LiDar-image rendering using NeRF techniques. The authors first analyze key limitations of the methods on pose-free NeRFs and LiDar-NeRFs, then address them by introducing several adaptive contributions constructing a unified LiDar-camera field and an adaptive and effective optimization pipeline. Experiments show the significant improvement on reconstruction metrics (expecially LiDar metrics).

**Questions:**

See Weaknesses.

**Ethical Concerns:**

["NO or VERY MINOR ethics concerns only"]

**Final Justification:**

The authors' rebuttal addresses my concerns very well. I'm glad to hold my original rating to borderline accept this paper. Presentation issues should be further considered upon the publication.

**Limitations:**

Yes.

**Paper Formatting Concerns:**

NA.

**Quality:**

3

**Strengths And Weaknesses:**

Strengths:

1. Clear motivation and good technical contribution: The paper proposes a unified neural field for both LiDar and image rendering, as well as an effective and adaptive optimization skill. Current solutions on LiDar-NeRF design do need improvements like that to address key limitations.
2. Good expression: In method part, the authors do necessary explainations and exhibitions on why the module is designed like this -- it's explainable, clear and experimentally effective.
3. Strong evaluation results: the proposed method gains impressive results against related baselines, especially on LiDar metrics becaused of the unified field design.

Weaknesses:

1. Format advice: The words in Fig. 1 are too small to be clear enough. The format for Table 2 and 3 seems squeezed.
2. Discussion and possible comparison with Gaussian Splatting counterparts: I'm not here to say that the technical contribution on NeRF filed is not as meaningful as previous. However, I believe the authors can put 1-2 baselines based on Gaussian representation to make the experimental results more persuasive. After all, the technical stream based on GS shows faster rendering and optimization time. I would recommend NopeNeRF like baselines like COLMAP-free GS, and Dust3R-inspired(-utilized) baselines like InstantGS for the setting of pose-free reconstruction. Moreover, GS-based methods considering LiDar inputs also should be referred to.

---

> ### Author Rebuttal · Authors · 2025-07-30
>
> We sincerely appreciate your careful review. Your insightful and objective suggestions have been thoroughly considered, and the manuscript will be revised accordingly, including further experiments to enhance the persuasiveness of our method.
>
> **1. Formatting Issues:**  We aimed to present more comprehensive content in the main text, which unfortunately resulted in some formatting congestion. In future revisions, we will strive to improve the clarity of our figures and overall presentation.
>
> **2. Comparison with Gaussian Splatting:** We fully agree with your perspective, and accordingly, we selected the KITTI-360 dataset to evaluate the performance of COLMAP-free GS (CF-GS[1]) and DUSt3R[2]/VGGT[3]-assisted 3DGS reconstruction. For the latter, we select InstantGS[4] as the baseline method.
> |Method|CD↓|F-score↑|PSNR↑|SSIM↑|LPIPS↓|RPE_t(cm)|RPE_r|ATE(m)|
> |-|-|-|-|-|-|-|-|-|
> |CF-GS[1]|-|-|20.13|0.36|0.511|53.223|9.412|0.353|
> |InstantGS[4]|-|-|21.01|0.66|0.477|36.223|3.329|0.312|
> |Ours|0.079|0.942|23.46|0.76|0.287|1.471|0.025|0.187|
>
> 2.1 Comparison with CF-GS[1]: Similar to Nope-NeRF, the CF-GS also utilizes DPT-predicted depth as a prior to initialize the local 3D Gaussians. As a result, it suffers from the same limitations. Specifically, DPT’s depth estimation in outdoor scenes is inaccurate, and the predicted depth maps suffer from inherent scale ambiguity. Although CF-GS performs well in indoor and object-level scenes, it introduces many erroneous floating points in autonomous driving scenarios. Moreover, some frames in long driving sequences exhibit misaligned camera poses, the pose estimation quality is clearly inferior to approaches aided by LiDAR point clouds, which can achieve centimeter-level precision. Consequently, while CF-GS performs slightly better than Nope-NeRF, it still falls short of the performance achieved by our method.
>
> 2.2 Regarding the DUSt3R-inspired baselines, we sincerely appreciate your constructive suggestions. The results of InstantGS are shown in the table above. Due to the lack of robustness in end-to-end pose estimation for outdoor scenes, outlier frames occasionally occur. Thus, suboptimal initialization leads to degraded reconstruction quality in certain sequences. **Due to rebuttal constraints disallowing images and external links, we will make these results available on our future project page.**
>
> Recently, VGGT[3] has attracted considerable attention and provides a convenient API that enables direct pose estimation and point cloud generation from images for training 3DGS. To further evaluate the capability of end-to-end pose estimation in autonomous driving scenarios, we attempted to use VGGT outputs as initialization and subsequently performed 3DGS-based reconstruction. We observed that, although VGGT yields better performance than DUSt3R, it still struggles with long sequences in outdoor scenarios. Moreover, VGGT currently does not support using shared camera intrinsics as prior information, resulting in inconsistent intrinsic estimates across frames. These two factors together lead to notable radial distortions, similar to those observed in DPT and other monocular depth estimation methods. Consequently, the final reconstructed scenes contain noticeable artifacts and a large number of floaters.
>
> **3. Completeness of Related Work:** To improve the completeness of our related work, we will also include recent GS-based novel view synthesis methods of LiDAR point cloud in the updated version of the paper, such as GS-Lidar[5], LiDAR-GS[6] and others. Once again, we sincerely thank you for your valuable suggestions, which will help make our paper more thorough and comprehensive.
>
> [1] Fu, Yang, et al. "Colmap-free 3d gaussian splatting." Proceedings of the IEEE/CVF Conference on Computer Vision and Pattern Recognition. 2024.
>
> [2] Wang, Shuzhe, et al. "Dust3r: Geometric 3d vision made easy." Proceedings of the IEEE/CVF Conference on Computer Vision and Pattern Recognition. 2024.
>
> [3] Wang, Jianyuan, et al. "Vggt: Visual geometry grounded transformer." Proceedings of the Computer Vision and Pattern Recognition Conference. 2025.
>
> [4] Fan, Zhiwen, et al. "InstantSplat: Sparse-view Gaussian Splatting in Seconds." arXiv preprint arXiv:2403.20309 (2024).
>
> [5] Jiang, Junzhe, et al. "Gs-lidar: Generating realistic lidar point clouds with panoramic gaussian splatting." arXiv preprint arXiv:2501.13971 (2025).
>
> [6] Chen, Qifeng, et al. "Lidar-gs: Real-time lidar re-simulation using gaussian splatting." arXiv preprint arXiv:2410.05111 (2024).

---

> ### Author Response · Authors · 2025-08-05
>
> Dear Reviewer RGqZ,
>
> We genuinely value your insights and suggestions throughout this review process. As the discussion period comes to an end, we sincerely hope to hear your thoughts on our responses. If there is anything we may have missed or not fully addressed, we would greatly appreciate the opportunity to clarify or further discuss.

---

> > ### Comment · Area_Chair_pwws · 2025-08-06
> > **Author-reviewer Discussion**
> >
> > Dear reviewer,
> >
> > The system shows that you have not yet posted a discussion with the authors. As per the review guidelines, we kindly ask you to evaluate the authors’ rebuttal to determine whether your concerns have been sufficiently addressed before submitting the Mandatory Acknowledgement. If they have, please confirm with the authors. Otherwise, feel free to share any remaining questions or concerns.
> >
> > Thank you for your time and valuable feedback.
> >
> > Your AC

---

> > ### Comment · Reviewer_RGqZ · 2025-08-07
> >
> > The authors' rebuttal addresses my concerns very well. I'm glad to hold my original rating to borderline accept this paper. Meanwhile, I kindly remind the authors to notice the presentation issues (Figures and Tables) upon the publication.

---

> ### Author Response · Authors · 2025-08-07
>
> We truly appreciate your feedback. Thankfully, the final version of the manuscript offers more space to elaborate on our results, allowing us to present our findings in a more detailed manner and enhance the overall layout. Thank you once again for your review—it has played a significant role in strengthening the clarity and impact of our work.

---

### Official Review · Reviewer_NasR · 2025-07-01

**Clarity:** 3
**Significance:** 3
**Originality:** 3
**Rating:** 5
**Confidence:** 5

**Summary:**

This paper presents a unified framework of pose-free multimodal LiDAR-camera novel view synthesis. The view features and LiDAR features are unified in the hash grid of instant NGP. A multimodal geometric optimizer and a multimodal-specific coarse-to-fine training approach are utilized to guide the optimization of poses and neural fields.

**Questions:**

- MVS methods, e.g., colmap or other network-based methods, should also be compared with.

- How does Nope-NeRF utilize multimodal inputs in your baseline setting?

- Some important methods should be cited and well discussed.
[1] MVSNeRF: Fast generalizable radiance field reconstruction from multi-view stereo.
[2] DBARF: Deep bundle-adjusting generalizable neural radiance fields.
[3] CRAYM: Neural field optimization via camera ray matching.
[4] Self-calibrating neural radiance fields.

**Ethical Concerns:**

["NO or VERY MINOR ethics concerns only"]

**Final Justification:**

All of my concerns are addressed, so I'm raising the score to 5 accept.

**Limitations:**

Yes.

**Paper Formatting Concerns:**

The layout of Table 2 and Table 3 is too compact

**Quality:**

3

**Strengths And Weaknesses:**

Strengths:
- This paper presents a novel continuous and pixel-wise depth supervision.
- The proposed method outperforms other methods on both reconstruction and view synthetic results.
- The proposed method can better recover the poses of input views.

Weaknesses:
- The limitations of the proposed method are not well discussed.
- The comparisons are not quite fair.

---

> ### Author Rebuttal · Authors · 2025-07-30
>
> We sincerely thank you for your careful review. Your suggestions have been very helpful in improving our work and have provided us with valuable insights. In fact, the concerns you raised were considered at the early stages of this work. Below, we address each of them in detail.
>
> **1. Limitations of MUP:**  We briefly discussed some limitations of our method in Appendix(L525-L529), and here we provide a more detailed explanation.
>
> In practical applications, the primary limitation of our approach lies in the reconstruction of dynamic objects in autonomous driving scenarios. Our current method is not specifically designed for 4D (i.e., dynamic) reconstruction. A commonly adopted strategy is to mask out dynamic objects for static scene reconstruction, and then perform separate reconstruction of the dynamic components. We consider this process more of an engineering issue, as it typically relies on prior knowledge or external cues to identify and mask dynamic objects—something that is feasible with existing techniques. In future work, we plan to extend our method to more explicitly address dynamic scene reconstruction.
>
> In addition, there is a minor limitation related to the global optimization in MMG. The efficiency of this step relies on the sequential nature of the data: each frame’s point cloud (along with its corresponding image) is connected to a few temporally adjacent frames. If this sequential structure is absent, constructing the graph would require connecting each frame to all others, which would increase computational complexity and potentially degrade pose optimization performance. Fortunately, datasets in autonomous driving scenarios are typically captured as continuous sequences, allowing our method to operate efficiently in such settings.
>
> **2. Fairness of Comparisons:** As few existing baselines address multi-modal pose-free scene reconstruction, most of the baselines we compare against follow a "register-then-reconstruct" paradigm. We fully understand your concern regarding the fairness of such comparisons. To the best of our knowledge, point cloud registration generally achieves higher accuracy than image-based registration. Therefore, we choose to adopt Colored-ICP, which incorporates both image and point cloud information to improve alignment accuracy. To further address your concern, we provide additional experiments in our response to the next question, including both COLMAP-assisted and network-assisted registration strategies.
>
> **3. Comparison with COLMAP-assisted and Network-assisted Methods:** We fully agree with your points. In the early stages of this work, we carefully considered these comparisons. The reason we did not adopt COLMAP for the main evaluation is that Colored-ICP typically achieves higher registration accuracy than COLMAP. This is primarily because ICP-based method leverages geometric information for registration, while COLMAP is mainly based on sparse feature matching from images. Given that our method also utilizes geometric information (i.e., LiDAR point clouds), we believe that comparing with Colored-ICP is a more fair choice.
>
> | Method                    | CD↓   | F-score↑    | PSNR↑ | SSIM↑ | LPIPS↓ | RPE_t(cm) | RPE_r(degree) | ATE(m)   |
> |---------------------------|-------|-------|--------|--------|--------|--------|--------|--------|
> | Colored-ICP-assisted      | 0.492 | 0.787 | 20.92  | 0.698  | 0.459  | 25.383 | 0.899  | 1.624 |
> | COLMAP-assisted           | 0.532 | 0.781 | 20.96  | 0.675  | 0.488  | 32.383 | 1.962  | 1.016 |
> | GeoTransformer-assisted   | 0.314 | 0.811 | 22.21  | 0.701  | 0.333  | 6.024  | 0.393  | 0.254 |
> | **Ours**                  | **0.079** | **0.942** | **23.46** | **0.759** | **0.287** | **1.471** | **0.025** | **0.187** |
>
> To further address your concerns, we have now included COLMAP-assisted registration results in the above table. As shown, Colored ICP consistently achieves better registration accuracy than COLMAP under our settings. And regarding learning-based registration methods, we note that point cloud-based approaches generally offer higher accuracy compared to image-only methods. Among them, GeoTransformer[1] is a representative and state-of-the-art method for point cloud registration. We therefore conduct additional experiments using GeoTransformer with official pretrained weights (trained on the KITTI dataset). The results are summarized in the table above. Although GeoTransformer achieves good registration results compared to other baselines, our method attains superior pose estimation accuracy, with translation error reduced to one-third and rotation error reduced to one-sixteenth of GeoTransformer’s, thereby yielding improved reconstruction outcomes.
>
> **4. Multimodal Inputs in Nope-NeRF:** We follow the official setup of Nope-NeRF, which, under the original RGB-only input setting, employs DPT to estimate a depth map for each image. This introduces geometric priors that enable pixel-wise depth constraints. However, due to the inherent scale ambiguity in monocular depth estimation, the depth maps require joint optimization of their scale and offset. For more details, please refer to Appendix L571-575.
>
> Therefore, although Nope-NeRF is nominally an RGB-only method, we consider it to effectively leverage multi-modal inputs by incorporating estimated depth as a geometric prior. To the best of our knowledge, existing pose-free reconstruction methods primarily focus on RGB-only inputs. Among them, Nope-NeRF and the recently proposed COLMAP-free Gaussian Splatting (CF-GS)[2] are the most relevant methods that utilize multi-modal information for pose-free rescontruction, such as RGB combined with depth. We supplement here a comparison with CF-GS to better contextualize our method within this line of work. The experimental results on KITTI-360 can be referred to in our response to Reviewer RGqZ.
>
> **5. Related Work to Be Discussed:** We sincerely appreciate your valuable suggestions. The works you mentioned are highly relevant, especially those concerning generalizable NeRFs and sparse-view reconstruction, which we have been closely following. We will include citations and further discussion of these works in the revised version.
>
> (1) Generalizable NeRFs and Sparse-View Reconstruction: MVSNeRF[3] and PixelNeRF[4] are representative methods in the domain of generalizable NeRFs. They incorporate strong priors to enhance performance under sparse-view settings. Specifically, MVSNeRF constructs a cost volume by warping 2D image features onto a plane sweep and extracts voxel features via 3D CNNs, the features of query points are then obtained from the corresponding voxel features. DBARF[5] is more closely related to our approach. It extends generalizable NeRFs to pose-free settings by integrating a pose and depth estimator into the IBRNet backbone. While these methods primarily focus on generalizable NeRFs in object-centric or indoor scenes, and indeed exhibit faster convergence and improved robustness to sparse views—their application in autonomous driving scenarios remains challenging. **Specifically, the complexity of driving environments calls for stronger priors (e.g., in feature extraction and pose estimation), whereas methods that generalize well to object-level or indoor reconstructions often fall short.**
>
> (2) Pose-free NeRFs:
> CRAYM[6] focuses on pose-free reconstruction, approaching the task through camera ray matching and offering a novel perspective. It also shares conceptual similarities with ray-based pose estimation methods such as Ray-Diffusion[7]. However, it is primarily applied to object-level reconstruction. Among these methods, SCNeRF[8] is most closely related to ours. It adopts an optimization-based approach, leveraging NeRF to learn scene geometry while incorporating photometric consistency constraints in conjunction with traditional geometric priors from classical self-calibration techniques. **While SCNeRF represents a notable step forward in pose-free NeRF methods, Nope-NeRF leverages geometric priors to attain superior performance, especially in outdoor environments, as evidenced by the results presented in its experiments. **Therefore, we adopt Nope-NeRF as our baseline.**
>
> All these classical works have inspired our approach and are closely related to our method. Once again, we are grateful for the insightful suggestions and will incorporate the corresponding citations and discussions in the updated version.
>
> [1] Qin, Zheng, et al. "Geotransformer: Fast and robust point cloud registration with geometric transformer." IEEE Transactions on Pattern Analysis and Machine Intelligence 45.8 (2023): 9806-9821.
>
> [2] Fu, Yang, et al. "Colmap-free 3d gaussian splatting." Proceedings of the IEEE/CVF Conference on Computer Vision and Pattern Recognition. 2024.
>
> [3] Chen, Anpei, et al. "Mvsnerf: Fast generalizable radiance field reconstruction from multi-view stereo." Proceedings of the IEEE/CVF international conference on computer vision. 2021.
>
> [4] Yu, Alex, et al. "pixelnerf: Neural radiance fields from one or few images." Proceedings of the IEEE/CVF conference on computer vision and pattern recognition. 2021.
>
> [5] Chen, Yu, and Gim Hee Lee. "Dbarf: Deep bundle-adjusting generalizable neural radiance fields." Proceedings of the IEEE/CVF Conference on Computer Vision and Pattern Recognition. 2023.
>
> [6] Lin, Liqiang, et al. "Craym: Neural field optimization via camera ray matching." Advances in Neural Information Processing Systems 37 (2024): 9950-9973.
>
> [7] Zhang, Jason Y., et al. "Cameras as rays: Pose estimation via ray diffusion." arXiv preprint arXiv:2402.14817 (2024).
>
> [8] Jeong, Yoonwoo, et al. "Self-calibrating neural radiance fields." Proceedings of the IEEE/CVF international conference on computer vision. 2021.

---

> ### Author Response · Authors · 2025-08-05
>
> Dear reviewer NasR,
>
> We truly appreciate your time and thoughtful review. We believe we have addressed your concerns as best as we can. As the author-reviewer discussion period is coming to an end, we sincerely look forward to hearing your thoughts. If there are still any unresolved issues, we would be more than happy to continue the discussion.

---

> > ### Comment · Area_Chair_pwws · 2025-08-06
> > **Author-reviewer Discussion**
> >
> > Dear reviewer,
> >
> > The system shows that you have not yet posted a discussion with the authors. As per the review guidelines, we kindly ask you to evaluate the authors’ rebuttal to determine whether your concerns have been sufficiently addressed before submitting the Mandatory Acknowledgement. If they have, please confirm with the authors. Otherwise, feel free to share any remaining questions or concerns.
> >
> > Thank you for your time and valuable feedback.
> >
> > Your AC

---

> > ### Comment · Reviewer_NasR · 2025-08-06
> >
> > I greatly appreciate the efforts and detailed responses made by the authors in this work and the rebuttal. All of my concerns are appropriately addressed, so I am raising the score to "accept".

---

> > > ### Author Response · Authors · 2025-08-06
> > >
> > > We are genuinely glad that our replies have helped address your concerns. Your constructive feedback has been invaluable in further improving our work, and we sincerely appreciate the time and effort you dedicated to reviewing both our paper and rebuttal.

---

### Official Review · Reviewer_2Vdy · 2025-07-02

**Clarity:** 3
**Significance:** 3
**Originality:** 3
**Rating:** 4
**Confidence:** 4

**Summary:**

The authors propose MUP (Multimodal Unified Pose-free) framework, based on NeRFs for multimodal (LiDAR and RGB images) 3d reconstruction. Key contributions include a multimodal-specific coarse-to-fine (MSC2F) training scheme that balances the very different convergence speeds of images and point clouds. They also propose a multimodal geometric optimizer that uses Chamfer distance with point-to-image photometric regularisation to give reliable gradients for the pose. They demonstrate results on static scene from KITTI-360 and NuScenes which outperform baselines reported in the paper.

**Questions:**

1. How does MUP behave on sequences with moving vehicles or pedestrians? Could MMG reject dynamic points automatically?
2. Please answer queries in weaknesses section as well.

**Ethical Concerns:**

["NO or VERY MINOR ethics concerns only"]

**Final Justification:**

I'm happy with the detailed responses to my questions, therefore raising the score to a 4

**Limitations:**

yes

**Quality:**

3

**Strengths And Weaknesses:**

Strengths
1. Single hash grid for both modalities helps with better feature fusion and is a more elegant design choice.
2. Use of Image Weighted Chamfer distance is neat, and provides better grounding for training the already ill-posed setting of being pose-free.
3. I appreciate the ablations mentioned in the tables, they lay out the contributions well for each component.



Weaknesses
1. In the experiments I would like to see comparisons with some more recent baselines such as GeoNLF[1] (LiDAR only), and few other approaches like NeurAD[2], SplatAD [3]which are multimodal but use poses. In table 2 and 3 there are comparisons where authors use GT poses and therefore a comparison with methods that do multi-modal 3D reconstruction albeit with poses is warranted and will strengthen the contribution of the paper.

2. I’m confused as to why only the front camera was used for experiments? I’d like to see consistency of results when all cameras are used.

3. I would like to know more about the limitations of MSC2F such as mis-balances, or  where MMG fails due to low image-LiDAR overlap.

4. Typically, the sensor calibration(LiDAR-RGB) on the AV has some errors. A sanity check would be to see if rendered depth and rgb images are perfectly aligned using your multi-modal training scheme.

[1] Xue, Weiyi, et al. "GeoNLF: Geometry guided Pose-Free Neural LiDAR Fields." Advances in Neural Information Processing Systems 37 (2024): 73672-73692.

[2] Tonderski, Adam, et al. "Neurad: Neural rendering for autonomous driving." Proceedings of the IEEE/CVF Conference on Computer Vision and Pattern Recognition. 2024.

[3] Hess, Georg, et al. "Splatad: Real-time lidar and camera rendering with 3d gaussian splatting for autonomous driving." Proceedings of the Computer Vision and Pattern Recognition Conference. 2025.

---

> ### Author Rebuttal · Authors · 2025-07-31
>
> Thank you very much for your insightful suggestion, which helps improve the completeness of our work. The issues you raised have been well addressed, and we provide detailed responses below.
>
> **1. More comparisons:**
> Based on your suggestion, we provide comparisons with GeoNLF and NeuRAD on the KITTI-360 dataset, utilizing all stereo camera inputs. We evaluate on five static scenes. Note that NeuRAD's CD computation contains an error (see GitHub issue) and omits normalization by chunk size, we report results based on our corrected implementation. NeuRAD models the two modalities separately without explicit cross-modal information fusion. As a result, under ground-truth poses, our method performs slightly better in static scenes, while showing a significant advantage in the pose-free setting. Besides, compared to GeoNLF, our method benefits from the incorporation of multimodal inputs and achieves improved registration accuracy in the pose-free setting.
>
> |Method|CD↓|PSNR↑|SSIM↑|LPIPS↓|RPEₜ(cm)↓|RPEᵣ(°)↓|ATE(m)↓|
> |-|-|-|-|-|-|-|-|
> |NeuRAD|0.099|24.56|0.820|0.205|-|-|-|
> |Ours(GT-pose)(all cameras)|0.073|24.82|0.833|0.210|-|-|-|
> |GeoNLF|0.088| -| -| -|2.001|0.063|0.225 |
> |Ours(pose-free)(all cameras)|0.076|24.02|0.771|0.269|1.489|0.023|0.181|
>
> **2. Results when all cameras are used:** Your feedback is greatly appreciated and has been carefully considered. Indeed, using multiple cameras is more aligned with real-world autonomous driving settings. In the KITTI-360, it contains two forward-facing stereo cameras (the fisheye is typically not used for reconstruction tasks). Moreover, the two forward-facing cameras in KITTI-360 have a high degree of overlap and observe very similar views. Therefore, we only adopted a single pinhole camera in our original experiments. To maintain consistency, we used a similar setting for NuScenes.
>
> Motivated by your suggestion, we additionally conduct experiments by incorporating all available pinhole cameras in both KITTI-360 and NuScenes. We assume that the relative extrinsics between the cameras and LiDAR are known, which is a practical assumption in autonomous driving scenarios. In both datasets, sensors are time-synchronized, and all settings—including optimization steps, hyperparameters, and test sets—remain unchanged. The only difference is a larger set of RGB images with known poses used for supervision. We retain the alternating training strategy and report the results as follows:
>
> |Method|Dataset|CD↓|PSNR↑|SSIM↑|LPIPS↓|RPEₜ(cm)↓|RPEᵣ(°)↓|ATE(m)↓|
> |-|-|-|-|-|-|-|-|-|
> |Single|NuScenes|0.810|20.83|0.699|0.585|4.058|0.101|0.176|
> |All| |0.806|22.31|0.721|0.486|3.001|0.099|0.173|
> |Single|KITTI-360|0.079|23.46|0.759|0.287|1.471|0.025|0.187|
> |All| |0.076|24.02|0.771|0.269|1.489|0.023|0.181|
>
> As expected, incorporating additional supervised views improves reconstruction quality without introducing significant computational overhead. In particular, on the NuScenes dataset, leveraging six camera viewpoints leads to a notable performance gain. This improvement can be attributed to two key factors: (1) denser RGB supervision enhances photometric consistency; and (2) LiDAR points are projected onto a greater number of views, enabling our MMG module to more effectively identify and filter out occluded or non-overlapping regions.
>
> **3. Potential Limitations of MSC2F and MMG under Low Image-LiDAR Overlap:**
>
> (1) Effect of Non-overlapping Fields of View (FoV) on MMG.
> Our setup utilizes a 360-degree surround-view LiDAR, and thus the primary source of non-overlap arises from the mismatch in vertical FoV between the LiDAR and the RGB cameras (LiDAR typically has a much narrower vertical FoV). Despite this limitation, the LiDAR still provides rich geometric cues that can be leveraged for accurate pose optimization. As demonstrated in Tab.2 (w/o image), although the absence of image supervision leads to a significant performance drop, pose optimization still converges to a reasonable range, thereby establishing a lower bound for MMG’s performance.
>
> (2) Optimization Balance in MSC2F.
> MSC2F employs a hard-coded scheduling strategy in which each modality is gradually and fully activated during training. **This ensures that both image and LiDAR features are eventually optimized, regardless of their degree of FoV overlap.** Consequently, even under low-overlap scenarios between RGB and LiDAR, each modality is independently capable of contributing to the reconstruction. **The primary source of imbalance lies in the frequency content: LiDAR point clouds are inherently sparser than images**, whereas RGB inputs provide richer high-frequency details. We find that if the LiDAR branch converges too quickly, the optimization process may shift its focus toward fine-grained texture details in RGB, potentially undermining the representation of geometry. While this modality imbalance does not lead to failure in LiDAR-based reconstruction (as shown in Tab.2,w/o MSC2F), it can degrade geometric fidelity. To address this, we propose the MSC2F training strategy, which ensures stable convergence across all sequences.
>
> Another implication of low-overlap conditions lies in the extrapolation of LiDAR-based geometry. As discussed in L306-308, thanks to the continuity of neural fields, our model can extrapolate depth estimates beyond the LiDAR’s limited FoV by leveraging supervision from overlapping images (see Fig. 6). Naturally, the extrapolation becomes less reliable if the LiDAR FoV is further reduced. Nonetheless, in the KITTI dataset, where the LiDAR has only approximately half the horizontal field of view compared to the camera, our method is still able to produce stable and accurate extrapolated depth.
>
> **3. Sensor Calibration concerns:**  We appreciate your insightful suggestion. Since the data in the dataset have been carefully processed and well-calibrated, such errors theoretically have minimal impact on the results, as shown in Fig.7(last row), where the depth maps and images are well-aligned. Furthermore, by combining the rendered depth with corresponding color information and back-projecting them into 3D space, we obtain dense colored point clouds that appear visually consistent and plausible.(**Due to rebuttal constraints disallowing images and external links, we will make these results available on our future project page.**)
>
> We fully understand your concern, which is indeed critical in real-world applications. To address this, while retaining the original relative poses as priors, we introduce a small, learnable adjustment by enabling gradient propagation to the relative poses, analogous to pose optimization. Given that the initial relative poses are reasonably accurate, this refinement is performed at a later training stage or with an extremely small learning rate to maintain stability throughout the optimization process.
>
> We conducted a simple experiment on a sequence in KITTI-360: the relative poses are initialized using the provided values, perturbed with an average angular error of 2 degrees, and optimized with a learning rate 20× smaller than that used for the extrinsic parameters. **The results are close to those obtained using the provided values, with a comparable Chamfer Distance, a PSNR drop of only 0.06, and a reduction in the relative rotation error from 2 degrees to 0.12 degree.**
>
> Nonetheless, in practical deployments where relative poses are not guaranteed to be accurate, our framework retains an interface for such refinements, providing flexibility to adapt to different levels of pose uncertainty.
>
> **4. Dynamic scenes:** This is a valuable question. Since our method does not explicitly model dynamic objects, the NVS results exhibit noticeable blurring in dynamic regions. We have acknowledged this limitation in Appendix (L525–529).
> Currently, most approaches handle dynamic scenes by separately reconstructing static and dynamic components—typically by masking out dynamic objects during static scene reconstruction and modeling dynamic objects independently. We consider this primarily an engineering aspect that requires introducing appropriate priors to mask dynamic elements, a task that has been extensively studied and is relatively straightforward to implement. Addressing this issue will be a key direction for future work to further improve our method.
>
> Regarding the second question, our response is: **Yes, MMG is effective at filtering out dynamic parts**, and even in the presence of dynamic objects, our pose optimization still achieves high accuracy. For instance, in the sequence 2350-2400 of KITTI-360 (which includes moving vehicles), our method successfully performs registration. This can be attributed to two main reasons:
>
> (1)In MMG, the CD is computed by finding the nearest neighbor $b \in B$ for each $a \in A$ after global alignment. We further project these points onto time-synchronized images and use the color differences to filter out non-overlapping regions (e.g., areas occluded in one frame but visible in the other). **This mechanism is also effective for dynamic objects because erroneous correspondences caused by motion are filtered out by MMG.** Concretely, if $a$ is a point on a static car (e.g., on a side mirror), its nearest neighbor $b$ in point cloud B after alignment will correspond to the same location on the car. Conversely, if $a$ belongs to a dynamic car, once aligned in the global coordinate system, its nearest neighbor in B is likely to be off the car (e.g., on a building), causing significant color differences that lead to filtering.
>
> (2)In autonomous driving scenarios, dynamic objects generally constitute a small portion of the overall point cloud, with the scene dominated by static geometry. Similar to how ICP can still register frames effectively in dynamic scenes given good initialization and sufficient overlap, our MMG optimization objective minimizes the CD over all edges in the graph, which is primarily governed by the static parts of the scene.

---

> ### Author Response · Authors · 2025-08-05
>
> Dear Reviewer 2Vdy,
>
> Thank you for taking the time to review our work and for your thoughtful and constructive comments. We sincerely hope that our rebuttal has addressed your concerns adequately. As the author-reviewer discussion period is nearing its end, we would greatly appreciate any further feedback you might be willing to share.
>
> Thank you again for your time and consideration.

---

> > ### Comment · Reviewer_2Vdy · 2025-08-05
> > **Update**
> >
> > Happy with the response, I'm raising my score

---

> > > ### Author Response · Authors · 2025-08-06
> > >
> > > We sincerely thank you for your thoughtful reconsideration. We are truly encouraged to know that our responses have helped clarify the concerns you raised. We deeply appreciate the time and effort you devoted to reviewing both our submission and rebuttal.

---

### Official Review · Reviewer_UVxv · 2025-07-02

**Clarity:** 2
**Significance:** 2
**Originality:** 2
**Rating:** 5
**Confidence:** 3

**Summary:**

This paper presents a multi-modal, unified framework for pose-free novel view synthesis tailored to autonomous driving scenarios. Unlike previous approaches that rely on a single modality, either RGB images or LiDAR point clouds. By integrating supervision from both RGB and LiDAR,  the proposed method improves pose optimization, depth estimation accuracy, and the overall quality of synthesized novel views.

**Questions:**

See weaknesses

**Ethical Concerns:**

["NO or VERY MINOR ethics concerns only"]

**Final Justification:**

Thanks for the rebuttal! The clarification of technical contribution can be included into the introduction part. I understand that due to the rebuttal policy of this year, authors cannot upload any visuals. But it would improve the paper if the authors can add GT images in the figure comparisons in the final revisiom. Also, it will help readers better understand Fig. 6 if you have sperate visuals of LIDAR and RGB inputs, network outputs, and GT with clear captions.

I went through the questions from other reviewers. The authors also provided additional comparisons and results to address their concerns.

**Limitations:**

Yes

**Paper Formatting Concerns:**

In Table 2, the vspace is too much.

**Quality:**

2

**Strengths And Weaknesses:**

Strengths:

The paper introduces a Multimodal Geometric Optimizer that jointly leverages geometry and RGB information to refine camera poses explicitly. The proposed rigid alignment constraints significantly improve pose estimation accuracy, as evidenced by the results in Table 2.

The method demonstrates superior performance in geometry estimation, novel view synthesis, and pose prediction on two challenging datasets: KITTI-360 and NuScenes.

Detailed ablation studies analyzed the efficacy of each design, like unified modality training, Multimodal-specific Coarse-to-fine Training, and Multimodal Geometric Optimizer.




Weaknesses:

The technical contributions appear to be incremental rather than fundamentally novel from an academic perspective. (1) i-NGP also unified LiDAR-Camera observations, but it used separate hash grids for two different modalities. This work used a shared hash grid for both modalities. This modification seems to be a minor architectural variation and builds directly upon established techniques.
(2) The multimodal-specific coarse-to-fine training is more like an effective training trick rather than a fundamentally new algorithmic contribution to modulate the contribution of geometric and photometric losses across scales.

Lack of Ground Truth in Figures: Figures 3, 4, 5, 6, 7 do not include ground truth (GT) renderings for reference. This omission makes it difficult to assess the visual accuracy or quality of the proposed method. Including GT in at least some visualizations would significantly improve interoperability.

Ambiguity in Figure 6: The purpose and interpretation of the last row in Figure 6 are unclear.

Missing Video Results: The paper only provides static novel view comparisons with a few selected viewpoints. Without video results of RGB and depth renderings, it's difficult to evaluate the method’s temporal coherence or robustness across continuous novel views.

Including video examples (e.g., in supplementary materials) would strengthen the experimental section.

Lengthy Related Work in Introduction: The introduction section spends a significant amount of space reviewing related work, which could be streamlined. A more concise overview would help improve focus and readability.

---

> ### Author Rebuttal · Authors · 2025-07-30
>
> Thank you very much for your careful reading and summary. Below, we provide our responses to your questions, and we sincerely look forward to engaging in in-depth discussions with you.
>
> **1. Technical Contributions:** We would like to offer some additional clarifications. Our core motivation is to leverage multimodal information to achieve mutual complementarity, both in reconstruction and registration.
>
> (1) We observe that using separate hash grids for each modality introduces geometric inconsistencies across modalities, as the density values at the same spatial location can diverge. Especially for the camera field, continuous geometric supervision from the LiDAR field becomes difficult due to modality-specific density representations.
> To mitigate this, we adopt a shared density field to enforce cross-modal geometric consistency. In this setting, the camera field can obtain better geometric information from the LiDAR field (the density curve along the ray is sharper). Conversely, the LiDAR field benefits from the camera field by exhibiting increased continuity (enhanced interpolation capability).
>
> (2) In addition, maintaining independent hash grids increases both the optimization complexity and the risk of geometric misalignment in pose-free scenarios. By contrast, a unified hash grid imposes shared structure across modalities and proves to be more effective. From a technical standpoint, the instability inherent to ill-conditioned optimization limits the complexity of the methods that can be effectively applied. Thus, it is crucial to reduce the optimization burden as much as possible.
>
> Consequently, using a shared hash grid naturally offers an efficient solution to the aforementioned two challenges. However, direct fusion of modalities may introduce conflicts. For instance, we observe that if the LiDAR branch converges too rapidly, the optimization tends to overemphasize fine-grained texture details in the RGB modality, potentially compromising the geometric representation. To mitigate this, we propose the MSC2F strategy — a simple yet effective approach that progressively activates modalities in a coarse-to-fine manner, thereby balancing their contributions during training.
>
> **2.Ground Truth in Figures and Vedio example:** Thank you for pointing out the incompleteness of our presentation. As we aimed to compare different methods within limited space, some visual results were not sufficiently shown.
> In fact, our reconstruction—especially in point clouds—shows minimal perceptual differences from the GT, with little blur or artifacts in the rendered images. The main discrepancy lies in textureless regions where the GT appears sharper.
> (**Due to rebuttal constraints this year (e.g., no links or images allowed), we will provide more detailed visualizations on the project webpage  that will be released in the future, and include GT images in the revised version to enhance clarity and persuasiveness.**)
>
> **3.Ambiguity in Figure 6:** We apologize for the lack of clarity in our explanation. As shown in the leftmost image of the last row in Fig. 6, the LiDAR field of view (FoV) is significantly smaller than that of the image, resulting in noticeable non-overlapping regions. Combined with the second row of Fig. 6, our intention was to illustrate that: thanks to the continuity of neural fields, **accurate and smooth depth can still be recovered in image-supervised regions beyond the LiDAR FoV.** This effectively extrapolates the LiDAR FoV and demonstrates a potential benefit of the unified framework. The rightmost image in the last row is intended to show the full input view from the current camera perspective. We will clarify the purpose of these images in the revised version.
>
> **4.Lengthy Related Work in Introduction:** Thank you very much for your suggestion. In our initial submission, we aimed to thoroughly present the motivation behind our work, which unfortunately came at the expense of readability and conciseness. In the revised version, we will refine the focus of our exposition and relocate some of the detailed discussions to the Related Work section to improve clarity and overall presentation.

---

> ### Author Response · Authors · 2025-08-05
>
> Dear Reviewer UVxv,
>
> We sincerely appreciate your thoughtful review. As the author-reviewer discussion period is drawing to a close, we would be truly grateful to receive any further feedback you might have. If there are still any concerns that remain unaddressed, we would be more than happy to continue the discussion.

---

> > ### Comment · Area_Chair_pwws · 2025-08-06
> > **Author-reviewer Discussion**
> >
> > Dear reviewer,
> >
> > The system shows that you have not yet posted a discussion with the authors. As per the review guidelines, we kindly ask you to evaluate the authors’ rebuttal to determine whether your concerns have been sufficiently addressed before submitting the Mandatory Acknowledgement. If they have, please confirm with the authors. Otherwise, feel free to share any remaining questions or concerns.
> >
> > Thank you for your time and valuable feedback.
> >
> > Your AC

---

> > > ### Comment · Reviewer_UVxv · 2025-08-06
> > >
> > > Thanks for the rebuttal! The clarification of technical contribution can be included into the introduction part. I understand that due to the rebuttal policy of this year, authors cannot upload any visuals. But it would improve the paper if the authors can add GT images in the figure comparisons in the final revisiom. Also, it will help readers better understand Fig. 6 if you have sperate visuals of LIDAR and RGB inputs, network outputs, and GT with clear captions.
> > >
> > > I went through the questions from other reviewers. The authors also provided additional comparisons and results to address their concerns.

---

> > > > ### Author Response · Authors · 2025-08-06
> > > >
> > > > Thank you very much for your valuable suggestions. We will incorporate ground truth (GT) images and point clouds as you recommended and plan to showcase videos on the project webpage in the future. We are also working on presenting the full pose convergence process, which we believe will be more engaging and convincing. Fortunately, the main text of the final manuscript provides more space to present results, which will help us offer richer outcomes. Accordingly, we will address these aspects in the revised version by including GT visualizations, providing clear explanations under Figure 6, and presenting qualitative results in the single-modality comparison experiments.
> > > >
> > > > Once again, we sincerely thank you for your feedback, which will make our paper more complete and persuasive.

---

### Comment · Area_Chair_pwws · 2025-08-03
**Reviewer-Author Discussion**

Dear Reviewers,

Discussion with authors open until August 6 (AoE).

Please review the rebuttal and post any remaining concerns or comments if you have.

Kind regards,

AC

---

### Note · Authors · 2025-08-13

We are greatly encouraged that our responses have effectively addressed all of the reviewers’ concerns and have been met with positive feedback. Below, we present a concise synthesis of the rebuttal process:

**Strengths:** From an experimental standpoint, all reviewers provided favorable evaluations of our results, commending both the performance achieved and the comprehensiveness of the experimental validation. From the perspective of technical contributions, three reviewers explicitly described our work as “novel” and “elegant”, further acknowledging that our method “addresses a key limitation.”

**Questions and Weaknesses:**
(1) Presentation. Several reviewers noted that the layout appeared somewhat crowded and recommended the inclusion of additional visual examples as well as more detailed explanations for Figure 6. We have addressed these concerns by optimizing the layout, and, fortuitously, the final version will benefit from an additional page, enabling a clearer and more effective presentation of these enhancements.

(2) Experiments. Reviewer feedback primarily focused on comparisons with other pose-free methods, such as COLMAP-assisted approaches and 3DGS-based methods. We conducted efficient supplementary experiments demonstrating that our multimodal design not only achieves superior pose refinement accuracy but also delivers substantial improvements in overall scene quality.

(3) Limitations. Reviewers encouraged a deeper discussion of the method’s limitations and potential failure cases, particularly in scenarios with low overlap between images and LiDAR point clouds. We expanded on these aspects, including limitations in dynamic scenes, and demonstrated through ablation studies that, even under low-overlap conditions, our method consistently maintains performance above a practical lower bound. Additionally, we provided detailed explanations of how MMG automatically rejects dynamic points and discussed the extensibility of our sensor calibration optimization.

In conclusion, all reviewers agreed that we had satisfactorily addressed their concerns. This process has allowed us to further refine our results and enhance the overall persuasiveness of the work. We sincerely thank the reviewers and the AC for their time, effort, and valuable feedback.

---

### Decision · Program_Chairs · 2025-09-17

**Decision:**

Accept (poster)

**Comment:**

The paper received unanimously positive reviews, consisting of two Borderline Accepts and two Accepts. It introduces a novel pose-free framework that effectively combines two modalities, leveraging a compact neural representation without the need for accurate initial poses. While reviewers initially raised concerns regarding the method's incremental nature, unaddressed limitations, and insufficient comparisons, the rebuttal and subsequent discussion successfully resolved the main technical issues. As a result, Reviewers 2Vdy and NasR raised their scores to a Borderline Accept and an Accept, respectively, while the other two reviewers maintained their initial positive ratings. The AC judges that a consensus for acceptance has been reached and strongly recommends that the authors incorporate all reviewer feedback into the final camera-ready version.